# Deregulation of Cholesterol Homeostasis by a Nuclear Hormone Receptor Crosstalk in Advanced Prostate Cancer

**DOI:** 10.3390/cancers14133110

**Published:** 2022-06-24

**Authors:** Nianxin Yang, Yatian Yang, Zenghong Huang, Hong-Wu Chen

**Affiliations:** 1Department of Biochemistry and Molecular Medicine, School of Medicine, University of California, Davis, Sacramento, CA 95817, USA; nxyang@ucdavis.edu (N.Y.); yatyang@ucdavis.edu (Y.Y.); znhhuang@ucdavis.edu (Z.H.); 2National Cancer Institute Designated Comprehensive Cancer Center, University of California, Davis, Sacramento, CA 95817, USA; 3Veterans Affairs Northern California Health Care System, Mather, CA 95655, USA

**Keywords:** nuclear receptors, RORγ, LXRs, SREBP2, antagonists, inverse agonists, agonists, cholesterol efflux

## Abstract

**Simple Summary:**

Metastatic castration-resistant prostate cancer (mCRPC) is one of the leading causes of cancer-related death. mCRPC tumors feature high intratumoral cholesterol levels. Our current study shows that the nuclear receptor RORγ plays crucial roles in the aberrant cholesterol homeostasis of mCRPC. RORγ simultaneously stimulates cholesterol biosynthesis and suppresses cholesterol efflux programs through a RORγ and LXR crosstalk. Our study also demonstrated that RORγ antagonists alone or in combination with cholesterol-lowering drug statins are effective in inhibiting mCRPC cell and tumor growth. Our findings revealed a mechanism underlying the elevated cholesterol levels in mCRPC and suggested a potential therapeutic strategy for mCRPC patients.

**Abstract:**

Metastatic castration-resistant prostate cancer (mCRPC) features high intratumoral cholesterol levels, due to aberrant regulation of cholesterol homeostasis. However, the underlying mechanisms are still poorly understood. The retinoid acid receptor-related orphan receptor gamma (RORγ), an attractive therapeutic target for cancer and autoimmune diseases, is strongly implicated in prostate cancer progression. We demonstrate in this study that in mCRPC cells and tumors, RORγ plays a crucial role in deregulation of cholesterol homeostasis. First, we found that RORγ activates the expression of key cholesterol biosynthesis proteins, including HMGCS1, HMGCR, and SQLE. Interestingly, we also found that RORγ inhibition induces cholesterol efflux gene program including ABCA1, ABCG1 and ApoA1. Our further studies revealed that liver X receptors (LXRα and LXRβ), the master regulators of cholesterol efflux pathway, mediate the function of RORγ in repression of cholesterol efflux. Finally, we demonstrated that RORγ antagonist in combination with statins has synergistic effect in killing mCRPC cells through blocking statin-induced feedback induction of cholesterol biosynthesis program and that the combination treatment also elicits stronger anti-tumor effects than either alone. Altogether, our work revealed that in mCRPC, RORγ contributes to aberrant cholesterol homeostasis by induction of cholesterol biosynthesis program and suppression of cholesterol efflux genes. Our findings support a therapeutic strategy of targeting RORγ alone or in combination with statin for effective treatment of mCRPC.

## 1. Introduction

Deregulated cholesterol homeostasis at stages such as synthesis, efflux, uptake, storage and metabolism is often associated with tumorigenesis and cancer progression [1,2,3]. Cholesterol is not only a crucial component of cell membrane but also a key regulator of cell signaling via its control of membrane fluidity and lipid rafts. Cholesterol is also the precursor of metabolites such as steroids, oxysteroids, bile acids, and certain vitamins. In advanced prostate cancer (PCa) such as metastatic castration-resistant PCa or mCRPC, tumors often feature high intratumoral cholesterol levels, due to aberrant regulation of cholesterol homeostasis [4,5]. Notably, studies have demonstrated that both cholesterol biosynthesis and efflux are reprogrammed in mCRPC, which may be major contributors to tumor growth and lethal progression [6,7,8]. Indeed, elevated expression of cholesterol biosynthesis rate-limiting enzymes such as SQLE have been strongly correlated with poor outcome of PCa [7,9,10]. Low or loss of expression of key efflux proteins such as ABCA1 and metabolic enzymes such as CYP27A1 are also associated with the disease progression [6,11]. Moreover, aggressive tumors often display elevated contents of cholesterol esters and specific cholesterol metabolites, which is associated with tumor growth, metastasis, and drug resistance [12,13,14]. Loss of expression and function of tumor suppressors such as Pten and p53 has been causally associated with aberrant tumor cholesterol homeostasis. Heightened signaling by PI3K-Akt and mTOR or the androgen receptor (AR) also contributes to cholesterol deregulation [4]. Despite some recent progress, mechanisms underlying aberrant cholesterol homeostasis in advanced PCa are still poorly understood.

Cholesterol homeostasis is regulated chiefly at transcriptional level by two major groups of transcription factors. Sterol regulatory element-binding protein 1 and 2, particularly SREBP2, plays a pivotal role in activating cellular cholesterol biosynthesis genes such as HMGCS1 and SQLE, in response to low level of cholesterol [1], whereas liver X receptors (LXRs), members of the nuclear receptor superfamily, regulate cholesterol efflux [2]. LXR target genes include major cholesterol efflux-related enzymes or transporters, including ABCA1, ABCG1 and ApoE [15,16,17].

Cholesterol-lowering drug statins have shown tumor-suppressing activities in preclinical models [18]. However, their clinical trials have not shown significant benefit to PCa patients [19], likely due to tumor’s feedback activities in cholesterol homeostasis, including SREBP2-mediated upregulation of cholesterol biosynthesis genes [19,20]. Therefore, therapeutics that suppress elevated intratumoral cholesterol levels without induction of the feedback response are optimal strategies. Recently, drugs or compounds such as ezetimibe that target NPC1L1 protein for blocking cholesterol absorption and terbinafine that can inhibit SQLE were shown to be effective in inhibition of PCa tumor growth [9,21], thus supporting the notion that targeting the tumor aberrant cholesterol homeostasis can be an attractive option.

RORγ and its immune cell-specific isoform RORγt, another member of the NR family, play important functions in control of tissue metabolism and immune response [3,22]. Recently, a number of antagonists/inverse agonists of RORγ have been developed and several of them are at clinical trials for autoimmune disorders [3,22,23,24]. Previous studies of us and others demonstrated that RORγ plays a crucial role in tumor growth and progression [3,25,26]. We also showed that small-molecule antagonists of RORγ such as XY018 and SR2211 are effective in blocking mCRPC cell and xenograft tumor growth [25]. In addition, our recent study identified RORγ as an essential activator of the entire cholesterol-biosynthesis program in triple negative breast cancer (TNBC), dominating the function of SREBP2 [26]. Here, we report that in mCRPC, RORγ acts both as an essential activator of cholesterol biosynthesis program and a major suppressor of cholesterol efflux program. RORγ antagonists effectively abolish statin-induced feedback regulation and inhibit tumor growth. Therefore, our findings suggest that RORγ is a new player of cholesterol homeostasis deregulation in PCa.

## 2. Materials and Methods

### 2.1. Cell Culture

C4-2B and 22Rv1 prostate cancer cells were cultured in RPMI-1640 medium supplemented with 10% FBS (Gemini or Hyclone). Cells were grown at 37 °C in 5% CO_2_ incubators. 22Rv1 cell line was obtained from American Type Culture Collection (ATCC), and C4-2B cell line was obtained from UroCor Inc. (Oklahoma City, OK, USA). Cells lines were tested being negative for mycoplasma regularly.

### 2.2. Chemicals

XY018 (Purity > 99%) was synthesized by WuXi AppTec (Austin, TX, USA). SR2211 (Purity > 98%) was obtained from TOCRIS (Bristol, UK). All statin and other compounds were obtained from Sigma-Aldrich (St. Louis, MO, USA), Selleck (Houston, TX, USA), or MedChemExpress (Monmouth Junction, NJ, USA).

### 2.3. Cell Viability and Growth Assays

For cell viability, cells were seeded in 96-well plates at 2000 cells per well in a total of 100 µL of media. Serially diluted small molecule compounds in 100 µL of media were added to each designated well after 24 h. After 4 days of treatment, culture media was removed and 50 µL of Cell-Titer GLO reagents (Promega, Madison, WI, USA) was added, and then luminescence was measured on Varioskan Lux multimode microplate reader (Thermo Fisher Scientific, Waltham, MA, USA). All experimental points were measured as triplicates, and the experiments were repeated at least two or three times. The luminescence of cells treated with vehicle was set at 100% viability, and all other data were standardized to percentage of viable cells.

### 2.4. Measurement of Cholesterol Content in Cells

22Rv1 cells were cultured in 6-well plates and treated with compounds for 48 or 72 h. After treatment, cells were digested off the plates by 0.5% trypsin in PBS and collected into 1.5 mL Eppendorf tubes. Cellular cholesterol was extracted as previously described [26]. Cholesterol level of each cell extract was measured with AmplexTM Red Cholesterol Assay Kit (Thermo Fisher Scientific, Waltham, MA, USA). Florescent readings were normalized to protein concentrations. All measurements were repeated three times, and the whole experiments were repeated for at least two or three times.

### 2.5. Cholesterol Rescue Assay

C4-2B and 22Rv1 cells were seeded in 6-well plates for 24h. The cells were randomly assigned into three groups, with each group treated with either vehicle (DMSO), or RORγ antagonists XY018 (5 µM) or SR2211 (5 µM) for 48 h. Each group of cells were also separated into three sub-groups, with each sub-group receiving cholesterol supplement (Sigma-Aldrich, C4951) in its designated quantity (0, 1.25, or 2.5 µg/mL medium). After treatment, all cells were collected and the cell numbers were counted with Countess 3 Automated Cell Counter (Thermo Fisher Scientific, Waltham, MA, USA) following the manufacturer’s instructions. All measurements were set as duplicates, and the whole experiments were repeated for at least three times.

### 2.6. qRT-PCR and Immunoblotting Analysis

Total RNA was isolated and purified from cells seeded in 6-well plates or from xenograft tumors. cDNA was reverse-transcribed using qScriptTM cDNA SuperMix (Quanta Biosciences, Beverly Hills, CA 95048, USA), and then amplified and measured with 2X SYBRGreen qPCR Mastermix (Bimake, Houston, TX B21202, USA) or PowerUpTM SYBRTM Green Master Mix (Thermo Fisher Scientific, Waltham, MA A25742, USA). The SYBR fluorescence values were collected, and the melting-curve was analyzed. Expression of each transcript was normalized by GAPDH as the internal reference, and expression change in folds was calculated. The experiments were performed at least two to three times, with internal duplicates or triplicates, and the data was presented either in heat maps or as mean values ± s.d. Sequences of the primers are listed in Appendix A. Cell lysates were analyzed by immunoblotting with antibodies recognizing indicated proteins. Details of the antibodies are listed in Appendix A.

### 2.7. RNA-seq and Data Analysis

22Rv1 cells were treated with vehicle, XY018 (1.25 or 5 µM), Simvastatin (1.25 or 5 µM), Atorvastatin (1.25 or 5 µM), or the combination of XY018 and each statin (1.25 µM each) for 48 h before RNA extraction. C4-2B cells were treated as previously described. RNA-seq library preparation, sequencing, and sequence data analysis were performed as previously described [26].

### 2.8. siRNA Transfection

siRNAs for gene knockdown were purchased from Thermo Fisher (Waltham, MA, USA; NR1H3 siRNA, s14685; NR1H2 siRNA, s19568) or Santa Cruz Biotechnology (Dallas, TX, USA; LXRα siRNA, sc-38828; LXRβ siRNA, sc-45316). Transfections were performed with DharmaFECT#1 (Dharmacon, Lafayette, CO, USA) or Lipofectamine^TM^ RNAiMAX (Invitrogen, Waltham, MA, USA) following the manufacturer’s instruction in OptiMEM (Invitrogen, Waltham, MA, USA). For RNA extraction, cells were treated with siRNAs for 48 h, and for protein extraction, cells were treated with siRNAs for 72 h.

### 2.9. Mouse Models and Treatments

Four-week-old male mice (strain: NOD.CB17-Prkdc^scid^/NCrHsd) were purchased from Envigo (Indianapolis, IN, USA). Mice were housed under standard conditions, under a 12 h light/12 h dark cycle. For 22Rv1 cell line-derived xenograft, 2 × 106 cells were suspended in a total of 100 µL PBS and Matrigel (1:1) mixture, and implanted subcutaneously into the dorsal flank on both sides of the mice. For xenograft tumor growth curve analysis, when the tumor volumes were approximately 50 mm^3^, mice were randomized and then administered with 100 µL of vehicle (intraperitoneally (i.p.), in 15% Cremophor EL, Calbiochem, 82.5% PBS, and 2.5% DMSO), RORγ antagonists XY018 (5 mg/kg, i.p., in 15% Cremophor EL, Calbiochem, 82.5% PBS, and 2.5% DMSO), simvastatin (25 mg/kg orally, in PBS), or a combination of XY018 and simvastatin (by their respective dose and administration method). Tumor volumes were monitored every three days by using calipers with volume calculated by using the equation: π/6 (length × width^2^). Body weight was also monitored during the treatment period. At the end of the study, mice were sacrificed, and tumors were dissected and weighed. For tumor gene expression analysis, when the tumor volumes reached approximately 200 mm^3^, mice were treated with vehicle (i.p.), XY018 (25 mg/kg, i.p.), simvastatin (25 mg/kg, orally), or a combination of XY018 and simvastatin (by their respective dose and administration method) for 7 days. At the end of the treatment period, mice were sacrificed, and tumors were collected and subjected to RNA extraction.

The animal procedures were approved by the Institutional Animal Care and Use Committee (IACUC) of the University of California, Davis.

### 2.10. Statistical Analysis

All statistical details of experiments are included in the figure legends or the specific Method sections. The data are presented as mean values ± s.d. Statistical analysis was performed using two tailed Student’s t tests to compare the means. *p* < 0.05 was considered to be statistically significant.

## 3. Results

### 3.1. RORγ Antagonists Inhibit mCRPC Cell Growth and Survival by Decreasing Intracellular Cholesterol Levels

Our previous studies showed that RORγ is overexpressed and plays a crucial role in mCRPC tumors [25,27]. As demonstrated in our previous studies [25,27,28], small molecule antagonists (also known as inverse agonists) of RORγ such as XY018 or SR2211 potently inhibited the growth and survival of mCRPC cells (Figure 1a,b). Given that elevated cholesterol levels promote PCa cell growth and survival, we thus investigated whether the effect of RORγ inhibition is linked to its potential function in control of the increased cholesterol level in mCRPC. Thus, we measured cellular cholesterol levels in 22Rv1 cells treated with RORγ antagonist XY018 using a commercially available assay kit. Consistent with the cell viability test, RORγ antagonist XY018 significantly reduced cellular cholesterol levels in 22Rv1 cells (Figure 1c). To examine whether the reduction in cellular cholesterol levels contributes to the growth inhibition effect of the RORγ antagonists, we performed a cholesterol rescue experiment. Indeed, the RORγ antagonist-induced growth inhibition was largely mitigated by the exogenous cholesterol supply in both mCRPC cells (Figure 1d,e and Appendix A). Therefore, these results suggest that RORγ antagonists inhibit mCRPC cell growth and survival at least in part through decreasing cellular cholesterol levels. They also suggest that RORγ plays an important role in control of high cholesterol levels in mCRPC.

### 3.2. RORγ Controls Expression of Key Cholesterol Biosynthesis Enzymes in mCRPC Cells

In TNBC, RORγ can function as a master activator of cholesterol biosynthesis [26]. To examine whether in mCRPC, RORγ plays a similar function, we treated 22Rv1 and C4-2B cells with antagonist XY018 and performed qRT-PCR to analyze the changes in expression of the 21 enzyme encoding genes in cholesterol biosynthesis program. The analysis demonstrated that over half of the cholesterol biosynthesis genes are significantly downregulated by RORγ antagonist XY018 in both cell lines, including the rate-limiting enzymes HMGCR and SQLE, which are often upregulated in PCa tumors (Figure 2a). To further examine the effect of RORγ function inhibition on gene expression, we treated 22Rv1 cells with a low dose and a high dose of XY018 and performed RNA-seq analysis. Again, we found that over half of the cholesterol biosynthesis genes are downregulated by RORγ antagonist XY018 in a dose-dependent manner (Figure 2b). Consistently, XY018 treatment significantly downregulated protein expression of key cholesterol biosynthesis enzymes in both cell lines, including HMGCS1, HMGCR, SQLE, and DHCR24 (Figure 2c,d). In addition, we examined the effect of RORγ inhibition in androgen-responsive prostate cancer cell LNCaP and found similar but less dramatic inhibition in the gene expression (Appendix A). To validate that RORγ functions to activate cholesterol biosynthesis program, we treated the CRPC cells with RORγ specific siRNAs or RORγ agonists SR0987 and LYC55716 [29,30]. Consistent with the results from the antagonists, siRNA treatments significantly downregulated protein expression of key cholesterol biosynthesis enzymes in 22RV1 cells (Appendix A), while RORγ agonist treatments enhanced the protein expression in both 22RV1 and C4-2B cells (Appendix A). Together, these results suggest that RORγ functions as a major activator of cholesterol biosynthesis program in mCRPC.

### 3.3. Inhibition of RORγ Stimulates Cholesterol Efflux Gene Program in mCRPC Cells

Aberrant cholesterol homeostasis in PCa involves both elevated biosynthesis and reduced efflux of cholesterol [26,31]. To further examine the potential role of RORγ in control of other aspects of cholesterol homeostasis, we analyzed expression changes in all cholesterol homeostasis-related genes in the RNA-seq data of 22Rv1. Surprisingly, we found that expression of cholesterol efflux-related genes such as ABCA1, ABCG1, ABCG8, APOA1, -A5, APOE, LRP1 and NPC2 is significantly elevated by RORγ antagonist XY018 in 22Rv1 cells (Figure 3a). The three ATP binding cassette transporters are directly responsible for cellular cholesterol efflux. APOA and APOE are crucial components of lipoproteins that are responsible for cholesterol packaging and transport. Upon comparing the RNA-seq data from the culture of two different mCRPC cell lines (22Rv1 and C4-2B) treated with antagonist XY018, we found that gene programs involved in reverse cholesterol transport and cholesterol efflux are among the most highly enriched in the 671 commonly upregulated transcripts by XY018 in both 22Rv1 and C4-2B cell lines (Figure 3b). Our qRT-PCR analysis confirmed that ABCA1, ABCG1, MYLIP, APOA5, APOE, LRP1 and NPC2 are strongly induced by the RORγ antagonist in the two mCRPC cells (Figure 3c). Given the key role played by ABCA1 and ABCG1 in cholesterol efflux in PCa tumors, we analyzed their protein expressions in both 22Rv1 and C4-2B cells and found that RORγ antagonist XY018 significantly increased their protein levels (Figure 3d,e). In addition, siRNA knockdown of RORγ also enhanced the expression of ABCG1 in 22RV1 cells (Appendix A), while treatments of cells with RORγ agonists SR0987 and LYC55716 inhibited the expression of both ABCA1 and ABCG1 in 22RV1 and C4-2B cells (Appendix A). Therefore, these results suggest that in PCa, RORγ functions to suppress the expression of cholesterol efflux program and that targeting of RORγ with the inhibitors induces the efflux gene expression, which likely contributes to the overall effect of the inhibitors in reduction in cellular cholesterol level in PCa tumor cells.

### 3.4. LXRs Mediate the Regulation of Cholesterol Efflux Program by RORγ in mCRPC

RORγ is a well characterized transcriptional activator [32,33], which does not readily explain the induction effect of its antagonist on the cholesterol efflux genes. On the other hand, LXRα and LXRβ (with gene name NR1H3 and NR1H2, respectively) are well-known master regulators of cholesterol efflux [15,16,17]. Thus, we examined whether the effect of RORγ antagonism on cholesterol efflux gene program is via LXRs. qRT-PCR analysis revealed that RORγ antagonist XY018 treatment significantly enhanced LXRβ/NR1H2 gene expression in both 22Rv1 and C4-2B cells while slightly increased LXRα/NR1H3 gene expression in both cell lines (Figure 4a,b). In addition, the expressions of both LXRs along with the cholesterol efflux gene program were significantly elevated by RORγ antagonist XY018 treatment in androgen responsive LNCaP cells (Appendix A). Consistently, both LXRα and LXRβ protein expression were elevated by XY018 treatment in the mCRPC cells (Figure 4c,d). Moreover, RORγ specific siRNA treatments enhanced LXRβ protein expression in 22RV1 cells (Appendix A), while RORγ agonists SR0987 and LYC55716 treatments inhibited both LXRα and LXRβ protein expression in 22RV1 and C4-2B cells (Appendix A). Next, to determine whether the RORγ antagonist effect on cholesterol efflux is through LXR, we treated the cells with RORγ antagonist XY018 and LXRα/β-specific siRNAs. LXRα and LXRβ siRNAs effectively knocked down the expression of their respective target proteins in both cell lines (Figure 4e,f). In 22Rv1 cells, single treatment of siLXRα and siLXRβ and their combination treatment were all effective to significantly abolish the induction of ABCA1 and ABCG1 genes by XY018 (Figure 4g,h). On the other hand, in C4-2B cells, single treatment of siLXRα and siLXRβ showed relatively moderate effect on the expression induction by XY018, while their combination treatment significantly abolished the expression induction (Figure 4i,j). Together, these results indicate that the function of RORγ in suppression of cholesterol efflux gene program is through its positive regulation of LXR expression in mCRPC cells.

### 3.5. RORγ Inhibition Synergizes with Statins in Killing mCRPC Cells through Abolishing Statin-Induced Feedback

Statins are widely used as cholesterol lowering drugs [34,35]. Despite promising results from preclinical studies, statins have not shown remarkable benefits to advanced PCa patients in clinical trials [19]. As shown in Figure 5a, treatment of mCRPC cells with a relatively low concentration (1.25 μM) of either RORγ antagonist XY018 or simvastatin did not elicit a significant decrease in cellular cholesterol level. However, their combination significantly reduced the cholesterol content (Figure 5a). Remarkably, when combined with XY018, the widely used statins such as simvastatin (SMV), atorvastatin (ATV), fluvastatin (FLV) and pitavastatin (PTV) all possessed prominent synergistic effect in growth inhibition on both CRPC cell lines, indicating that the growth inhibition synergy with the RORγ antagonist is not limited to a specific statin drug (Figure 5b,c and Appendix A).

Induction by statin of a feedback, up-regulation of cholesterol biosynthesis program and consequently rebound of tumor cholesterol level is postulated to be a major reason underlying the lack of efficacy at the clinic [1,19]. Thus, to examine whether statin elicits a similar feedback mechanism in mCRPC cells, we performed RNA-seq analysis of 22Rv1 cells treated with simvastatin and atorvastatin. Our gene ontology analysis of the altered gene expression revealed that among the 1885 transcripts upregulated by both statins (Appendix A), cholesterol biosynthesis-related programs are among the most highly enriched (Figure 3d). To examine the impact of RORγ antagonists on statin-induced feedback, we performed RNA-seq profiling of cells treated by both simvastatin and XY018. Comparing the altered gene expressions in cells treated by the statin alone, XY018 alone or their combination revealed a significant overlap of 222 genes that are up-regulated by statin and down-regulated by either XY018 alone or XY018 in combination with simvastatin (Figure 5d). Gene ontology analysis of the 222 genes showed that cholesterol and isoprenoid biosynthetic processes are the most significantly altered gene programs (Figure 5e), therefore indicating that when used in combination, RORγ antagonist can strongly mitigate the feedback induction of cholesterol biosynthesis programs by statin. Indeed, RNA-seq and qRT-PCR analysis demonstrated that treating cells with the RORγ antagonist not only abolished the statin induction of cholesterol biosynthesis genes but also resulted in a net decrease in the gene program (Figure 5f,g). Remarkably, the combination treatment not only abolished statin-induced increase in cholesterol biosynthesis enzyme proteins, such as HMGCS1, HMGCR, SQLE, and DHCR24, but also resulted in a net decrease in their expression in both 22Rv1 and C4-2B cell lines (Figure 5h,i). Together, these results suggest that RORγ inhibition can synergize with statins in killing mCRPC cells and that one mechanism is that the inhibition effectively abolishes statin-induced feedback induction of cholesterol biosynthesis program.

### 3.6. Targeting RORγ in Combination with Statin Strongly Inhibits mCRPC Tumor Growth through Reprogramming Cholesterol Homeostasis

Knowing that RORγ antagonists and statins have synergistic effect in killing mCRPC cells, we next evaluated the therapeutic potential of the combination treatment in 22Rv1 xenograft tumor model. Intraperitoneal administration of a relatively low dose (5 mg/kg) of RORγ antagonist XY018 alone or oral administration of simvastatin (25 mg/kg) alone significantly inhibited the tumor growth by around 40% in tumor size. Notably, their combined treatment showed a significantly stronger tumor inhibition than either alone (Figure 6a). Tumor weights were also measured and were consistent with tumor sizes (Figure 6b). Moreover, no significant change in the animal body weight was observed over the course of the treatment (Figure 6c). Consistent with the results from 22Rv1 cells, tumor cholesterol biosynthesis gene expression was significantly inhibited by XY018 and its combination with statin. Importantly, tumor cholesterol efflux genes such as ABCA1, ABCG1 and the master regulator LXRs were also induced by XY018 treatment (Figure 6d,e). Together, these results suggest that RORγ antagonist alone or its combination with statin can be effective in inhibition of mCRPC tumor growth and that down-regulation of cholesterol biosynthesis program and up-regulation of cholesterol efflux genes are the underlying mechanisms.

## 4. Discussion

It is well established that tumors of advanced PCa feature a highly elevated cholesterol content. Previous studies have focused on the deregulation of cholesterol biosynthesis pathway and revealed aberrant expression and function of key enzymes such as SQLE. However, the mechanisms of how the different aspects of cholesterol homeostasis such as cholesterol efflux and biosynthesis are coordinately deregulated in the tumor are much less understood. In this study, we demonstrated that RORγ not only activates the expression of cholesterol biosynthesis program but also suppresses the expression of key cholesterol efflux genes, which include the major transporters such as ABCA1 and ABCG1 and APOA1. Therefore, our study identified, for the first time, RORγ as a unique transcriptional regulator that coordinately de-regulates the programs of cholesterol biosynthesis and efflux.

RORγ was identified to play a major role in PCa due to its prominent function in activating AR gene expression and enhancing AR function in driving PCa progression [25]. Later studies by us and others showed that RORγ also plays important roles in breast cancer, pancreatic cancer and small cell lung cancer through stimulating gene programs of metabolism, cancer stemness, proliferation, EMT, drug resistance and lineage fate [26,36,37]. Like its T cell isoform RORγt, tumor cell RORγ acts primarily as a potent transcriptional activator. Indeed, in TNBC, RORγ interacts with SREBP2 to hyper-stimulate cholesterol biosynthesis program. In this study, we observed a similar function of RORγ in up-regulation of cholesterol biosynthesis, including the effective abolishment of statin-induced, SREBP2-mediated feedback mechanism by the antagonist of RORγ. Therefore, it is likely that a mechanism similar to the one in TNBC is responsible for the RORγ function in control of PCa cholesterol biosynthesis program. However, interestingly, in PCa, we observe that RORγ suppresses the expression of LXRα and β, the two master regulators of cholesterol efflux program, which then leads to the suppression of the efflux genes. This is rather unique in that instead of acting as an activator, RORγ acts as a repressor to silence the two LXR genes. It is also uncommon that a member of the NR family such as RORγ controls the expression of the other NR members. Nevertheless, these observations underscore the central role played by RORγ in control of cholesterol homeostasis in advanced PCa. Currently, it is unclear how RORγ acts to silence LXR genes. Elucidation of the mechanism will likely take integrated approaches to identify the binding site of RORγ at the LXR genes and the co-factors involved.

Although both RORs and LXRs are considered as sensors of specific lipids, LXRs generally play a tumor-suppressive role in several types of cancer, including prostate cancer [3,15,38]. Although our current study is focused on the role of RORγ in control of LXR-mediated cholesterol efflux program, it is possible that their crosstalk may also occur at other pathways such as cell cycle, apoptosis, and oncogenic kinase signaling which appear to be the targets of LXR agonists [38]. Interestingly, RORγ and LXRs are both major players in tumor immune microenvironment. LXRs are key regulators of the functions of macrophage and other tumor-infiltrated immune cells such as myeloid-derived suppressor cells (MDSCs). Therefore, future studies are warranted to further dissect the interplays between RORγ and LXRs in tumor cells and tumor microenvironment.

In PCa, many studies made the link of statin use to a reduced risk of disease progression such as PSA-based biochemical recurrence (BCR) and poor survival. However, clinical trials with statins have all failed to demonstrate a strong efficacy in treating advanced prostate cancer. Many factors likely contribute to the failure, which include statin-induced feedback activities of cholesterol homeostasis in the tumor, inter- and intra-tumor heterogeneity, lack of efficacy-indicating biomarker, and clear understanding of lipid metabolism in the disease including the impact of circulating cholesterol such as hypercholesterolemia and other lipids on the tumor [4,39,40,41]. In our current and previous studies, we found that antagonists of RORγ can potently abolish the statin-induced, feedback up-regulation of cholesterol biosynthesis program, thus suggesting that targeting RORγ can be a better therapeutic strategy. However, we recognize that the significance of our studies is limited by the pre-clinical models we used. Further work with more clinically relevant models and clinical studies are needed to address the limitations and to further support the rationale of targeting RORγ and to provide new insights into the role of RORγ in control of lipid metabolism in the tumor cells, the tumor microenvironment and the host.

Elevated cholesterol biosynthesis likely stimulates the disease progression through several means, including androgen production to sustain AR activation and function [42,43,44,45,46]. Recent studies have identified several cholesterol biosynthesis intermediates and cholesterol metabolites as RORγ agonists [47,48,49]. It is thus tempting to speculate that RORγ induction of elevated cholesterol levels can further enhance its own functions in a feedforward manner for hyper-activating AR and RORγ itself in driving the disease progression. Therefore, development of new antagonists of RORγ that are highly effective, either alone or in combination with other therapeutics such as statins or ezetimibe will be an attractive strategy for treatment of mCRPC.

## 5. Conclusions

In conclusion, our study identified RORγ as a crucial contributor to the aberrant cholesterol levels in advanced prostate cancer. We demonstrated that RORγ simultaneously enhances cholesterol biosynthesis program and suppresses cholesterol efflux program, resulting in elevated cholesterol levels and tumor growth. Interestingly, we found that RORγ mediates cholesterol efflux program by repressing LXR expression and function. We further demonstrated that RORγ antagonists possess significant synergism with statin in inhibition of mCRPC cell and tumor growth.

## Figures and Tables

**Figure 1 cancers-14-03110-f001:**
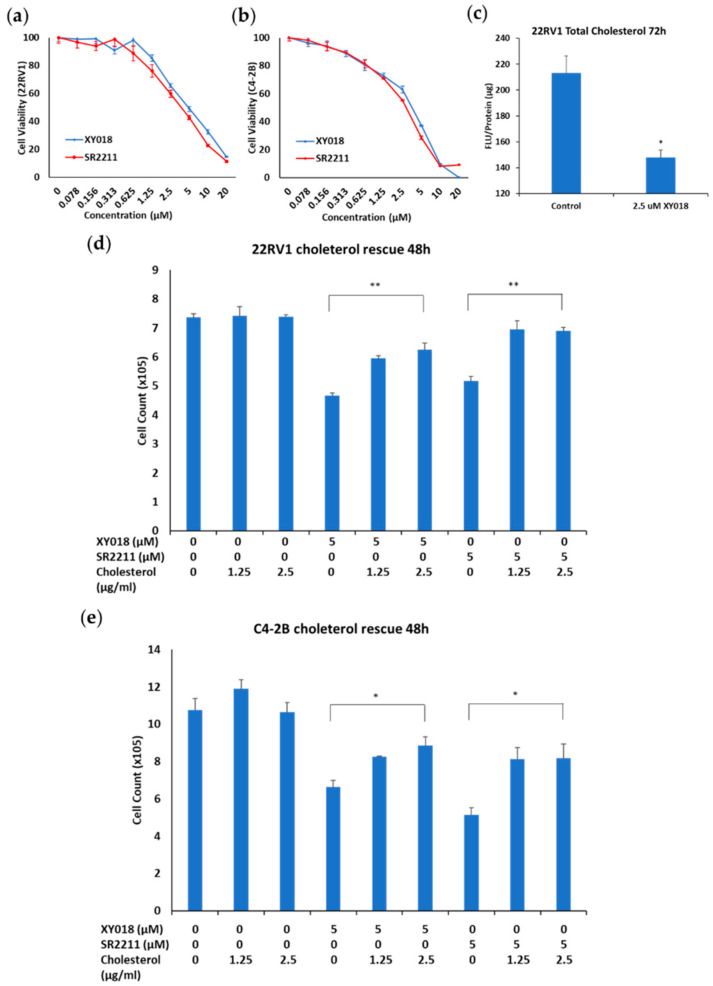
RORγ antagonists inhibit mCRPC cell growth and survival by decreasing intracellular cholesterol levels. (**a**,**b**) Cell viability, measured by Cell-Titer GLO (Promega) of 22Rv1 (**a**) and C4-2B (**b**) cells treated with the indicated concentration of RORγ antagonists XY018 and SR2211 for 4 d. (**c**) Total cholesterol levels in relative florescent units/protein, measured by Amplex^TM^ Red Cholesterol Assay Kit of 22Rv1 cells treated with indicated concentration of XY018 for 72 h. (**d**,**e**) Cell numbers of 22Rv1 (**d**) and C4-2B (**e**) cells treated with indicated concentration of XY018, SR2211, and cholesterol for 48 h. Data are shown as mean ± s.d. *n* = 3. Student’s *t* test. * *p* < 0.05, ** *p* < 0.01.

**Figure 2 cancers-14-03110-f002:**
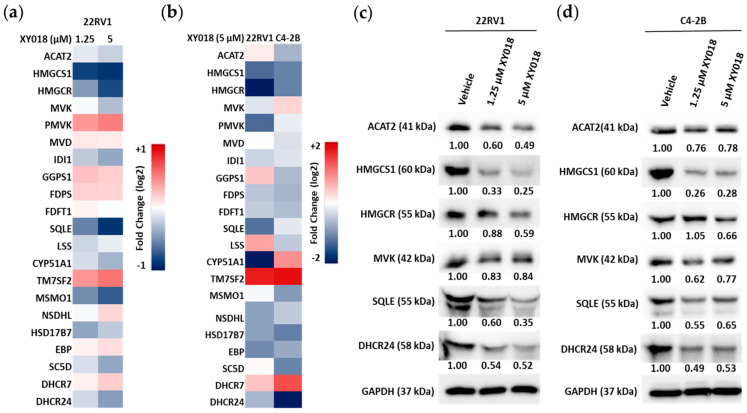
RORγ antagonists inhibit key cholesterol biosynthesis gene expression in mCRPC cells. (**a**) Heat map of mRNA expression of 21 cholesterol biosynthesis genes, as measured by qRT-PCR in 22Rv1 and C4-2B cells treated with 5 µM of XY018 for 48 h, as compared to vehicle (DMSO), *n* = 3. (**b**) Heat map of mRNA expression changes in 21 cholesterol biosynthesis genes, as detected by RNA-seq in 22Rv1 cells treated with indicated concentrations of XY018 for 48 h, as compared to vehicle (DMSO). (**c**,**d**) Immunoblotting of proteins involved in cholesterol biosynthesis pathway in 22Rv1 (**c**) and C4-2B (**d**) cells treated with indicated concentrations of XY018 for 72 h.

**Figure 3 cancers-14-03110-f003:**
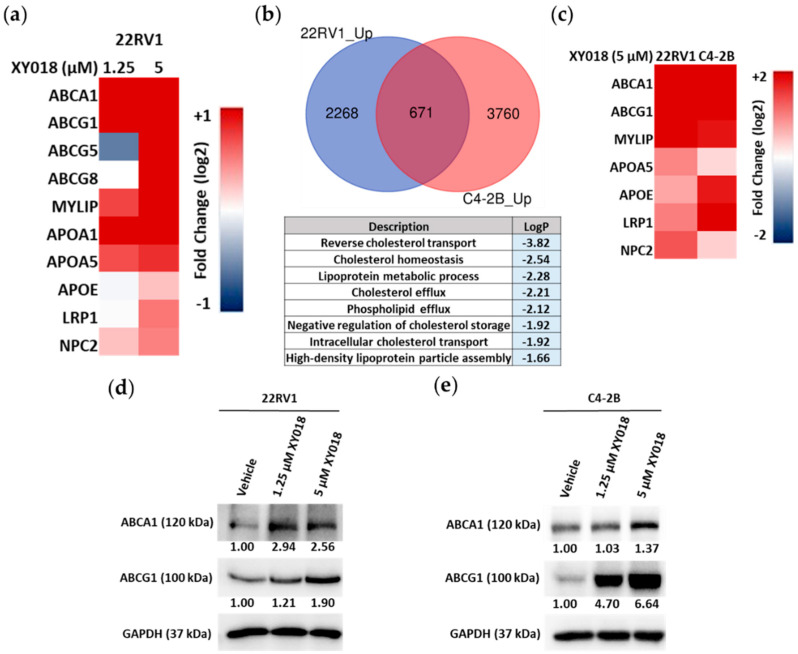
RORγ antagonists stimulate cholesterol efflux gene expression in mCRPC cells. (**a**) Heat map of mRNA expression changes in 10 cholesterol efflux genes, as detected by RNA-seq in 22Rv1 cells treated with indicated concentrations of XY018 for 48 h, as compared to vehicle (DMSO). (**b**) Venn diagram of the number of genes with expression significantly upregulated (1.3-fold), as detected by RNA-seq of 22Rv1 and C4-2B cells treated with 5 µM of XY018 for 48 h (top). Gene ontology analysis of the 671 genes with expression upregulated in both 22Rv1 and C4-2B cells treated with XY018 as shown in the top part (bottom). (**c**) Heat map of mRNA expression changes in 7 cholesterol efflux genes, as measured by qRT-PCR in 22Rv1 and C4-2B cells treated with 5 µM of XY018 for 48 h, as compared to vehicle (DMSO), *n* = 3. (**d**,**e**) Immunoblotting of proteins involved in cholesterol efflux pathway in 22Rv1 (**d**) and C4-2B (**e**) cells treated with indicated concentrations of XY018 for 72 h.

**Figure 4 cancers-14-03110-f004:**
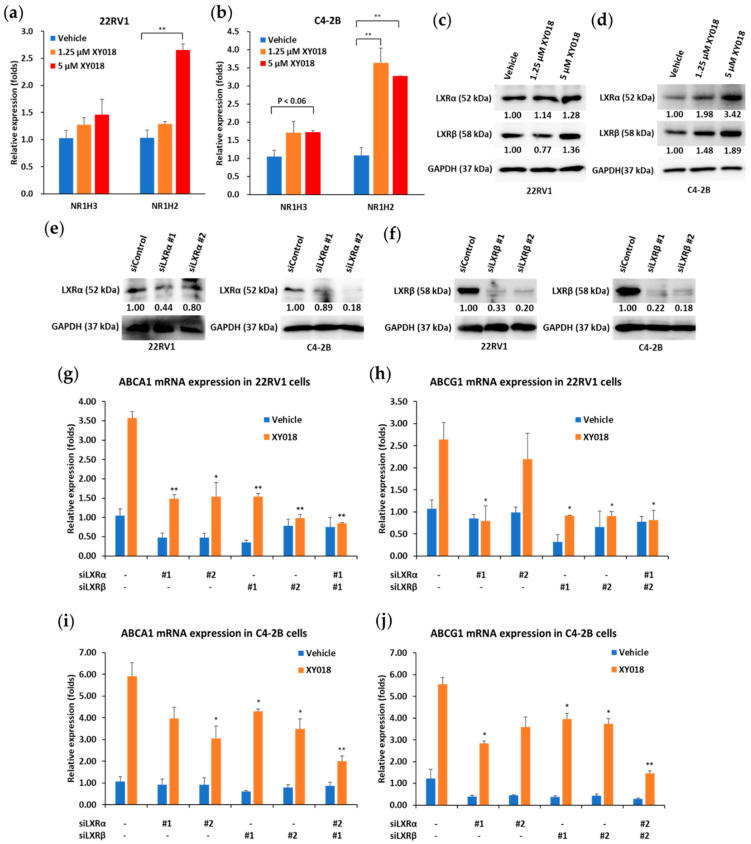
LXR expression regulates aberrant cholesterol efflux gene program mediated by RORγ in mCRPC cells. (**a**,**b**) Relative expression of NR1H3 and NR1H2 mRNA in fold changes, as detected by qRT-PCR in 22Rv1 (**a**) and C4-2B (**b**) cells treated with indicated concentrations of XY018 for 48 h. (**c**,**d**) Immunoblotting of LXRα and LXRβ proteins in 22Rv1 (**c**) and C4-2B (**d**) cells treated with indicated concentrations of XY018 for 72 h. (**e**,**f**) Immunoblotting of LXRα (**e**) and LXRβ (**f**) proteins in 22Rv1 and C4-2B cells treated with indicated siRNAs for 48 h. (**g**–**j**) Relative expression of ABCA1 and ABCG1 mRNA in fold changes, as detected by qRT-PCR in 22Rv1 (**g**,**h**) and C4-2B (**i**,**j**) cells treated with vehicle (DMSO) or 2.5 µM XY018 and indicated siRNAs for 48 h. The experiments were repeated three times. Data are shown as mean ± s.d. n = 3. Student’s *t* test. * *p* < 0.05 ** *p* < 0.01.

**Figure 5 cancers-14-03110-f005:**
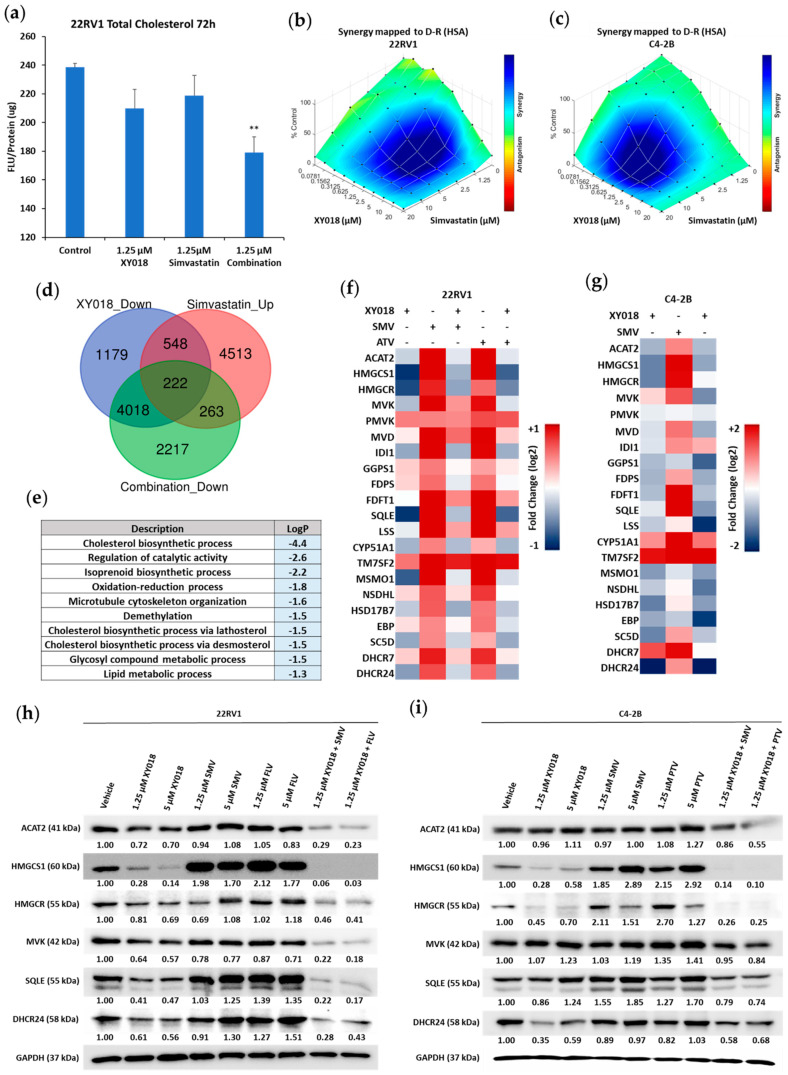
RORγ antagonists possess synergism with statins in killing mCRPC cells and abolish statin-induced feedback activities in cholesterol biosynthesis gene program. (**a**) Total cholesterol levels in relative florescent units/protein, measured by Amplex^TM^ Red Cholesterol Assay Kit of 22Rv1 cells treated with indicated concentration of XY018, Simvastatin, or their combination for 72 h. (**b**,**c**) Drug combination synergism maps of 22Rv1 (**b**) and C4-2B (**c**) cells treated with XY018 and SMV as indicated concentration for 4 d. Blue indicates synergy, while red indicates antagonism between drugs. (**d**) Venn diagram of the number of genes with expression significantly downregulated by XY018 (5 µM), or upregulated by SMV (5 µM), or downregulated by XY018 + SMV combination in 22Rv1 cells treated for 48 h, which are detected by RNA-seq. as detected by RNA-seq of 22Rv1 and C4-2B cells treated with 5 µM of XY018 for 48 h (top). (**e**) Gene ontology analysis of the 222 genes overlapped in expression alterations as shown in (**d**) in response to indicated compound treatment. (**f**) Heat map of mRNA expression changes in 21 cholesterol biosynthesis genes, as detected by RNA-seq in 22Rv1 cells treated with indicated concentrations of XY018 (5 µM), SMV (5 µM), ATV (5 µM), or XY018 + SMV/ATV combination (1.25 µM) for 48 h, as compared to vehicle (DMSO). (**g**) Heat map of mRNA expression changes in 21 cholesterol biosynthesis genes, as measured by qRT-PCR in C4-2B cells treated with XY018 (5 µM), SMV (5 µM), or XY018 + SMV combination (1.25 µM) for 48 h, as compared to vehicle (DMSO), *n* = 3. (**h**,**i**) Immunoblotting of proteins involved in cholesterol biosynthesis pathway in 22Rv1 (**h**) and C4-2B (**i**) cells treated with indicated concentrations of XY018, SMV, FLV, PTV or combinations of XY018 + statins for 72 h. Data are shown as mean ± s.d. *n* = 3. Student’s *t* test. ** *p* < 0.01.

**Figure 6 cancers-14-03110-f006:**
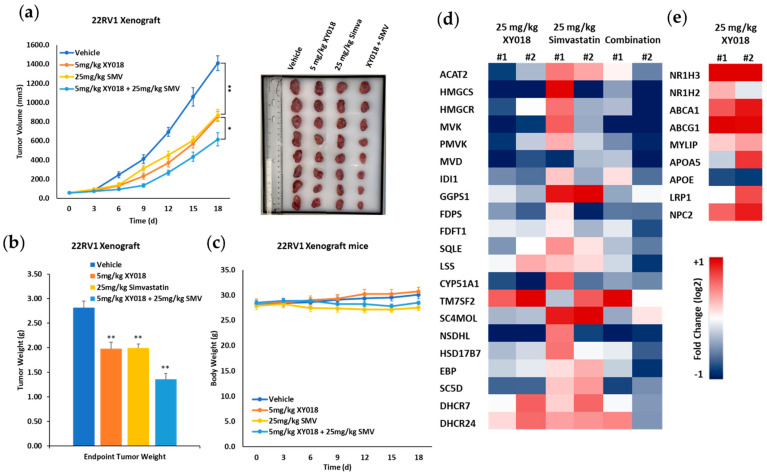
RORγ antagonist in combination with statin inhibit mCRPC tumor growth. (**a**) Growth inhibition effect of the indicated treatments (XY018, 5 mg/kg i.p. daily; SMV, 25 mg/kg orally daily; the combination of the two treatments; or vehicle) on 22Rv1 xenograft tumors (*n* = 7 mice per group). Representative images are shown. (**b**) 22Rv1 xenograft tumors were dissected and weighed at the end point of growth inhibition experiments from (**a**). (**c**) Mice bearing the xenograft tumors were weighed at the end point of the growth inhibition experiments from (**a**). (**d**,**e**) Heat maps of mRNA expression changes in 21 cholesterol biosynthesis genes (**d**) and 9 cholesterol efflux genes (**e**) as measured by qRT-PCR in the xenograft tumors treated with XY018 (25 mg/kg i.p. daily), SMV (25 mg/kg orally, daily), or XY018 + SMV combination for 7 d, as compared to vehicle. Data are shown as mean ± s.d. Student’s *t* test. * *p* < 0.05 ** *p* < 0.01.

## Data Availability

The data used to support this research are available from the corresponding author upon request.

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
