# Peer review of "Deregulation of Cholesterol Homeostasis by a Nuclear Hormone Receptor Crosstalk in Advanced Prostate Cancer"

_cancers, 2022, doi:10.3390/cancers14133110_

Round 1

Reviewer 1 Report

The authors answered all comments and suggestions.

Reviewer 2 Report

Authors addresses all comments from first report.

This manuscript is a resubmission of an earlier submission. The following is a list of the peer review reports and author responses from that submission.

Round 1

Reviewer 1 Report

COMMENT TO AUTHORS

This study aims to investigate the association between cholesterol homeostasis and advanced prostate cancer. The authors showed that the nuclear receptor RORγ plays crucial roles in the aberrant cholesterol homeostasis, , resulting in elevated cellular cholesterol levels in mCRPC.They concluded that RORγ antagonists could have a potential role for mCRPC patients.

I believe that the study has sufficient merit to be considered for publication on Cancers, although major revisions are required.

MAJOR COMMENTS

It could be confusing for the readers that authors performed some experiments  with two agonists in two cell lines and others only with an antagonist or only in a cell line.  Authors should motivate the rationale of their choices.

To support their hypothesis experiments should be performed also in an androgen responsive cell lines, as LNCaP.

In figure 1a and b, statistical analysis is lacking.

To assess the effect on cell growth and survival authors should perform a cell viability test other than cell count.In figure 1 authors used 1,25 and 2,5 uM of each agonist while in figure 2 they used 1,25 and 5uM.

To support their hypothesis experiments should be conducted also by silencing ROR gamma, or if exists with a ROR gamma agonist

- Keywords: remove CRPC and add advanced prostate cancer. - Introduction. I suggest providing a more detailed presentation of prostate cancer, in this regard, discuss this interesting paper (doi10.2144/fsoa-2020-0154; PMID 33552540). In addition , it will be for the benefit of the reader if the author focus more on the role of lipid metabolism and discuss how enhanced synthesis or uptake of lipids contributes to cancer cell growth and tumour formation (doi10.3390/diagnostics12020431 ; PMID 35204522). - There are several drugs used to treat cholesterol metabolism. What is the reason for which statins or ezetimibe are considered? - Please add the description of the acronyms ( e.g SQLE) and correct some typing errors (e.g. In line 257 “ cholesterol” and in line 277 the sentence is in different size). - Discussion. Could this study have implications for the relationship between hypercholesterolemia and prostate cancer? The authors should argue more about the discussion,  including the limitations of the study.

Reviewer 2 Report

Nianxin Yang et al present in their article results of study on retinoid acid receptor-related orphan receptor gamma (RORγ)and it's impact on  deregulation of cholesterol homeostasis in Advanced Prostate Cancer. The subject of the work is very interesting, especially since the analyzed aspects are not fully understood. 

Before considering publication, the article requires major revisions.

General comments:

The subject is of interest to the field.

The methods are correctly described and applied. 

The conclusions are consistent with the results.

References are adequate in number and range.

Major comments:

 - Abstract - should be structured in classic way - in present form is too descriptive like discussion not abstract  - needs thorough revision.
 - Results  - the raw results obtained in the analyzes should be presented - the descriptive fragments should be transferred to the discussion

 - Discussion  - definitely too short - since most of the information is in the Results section - it definitely needs a redesign.

 - and at the end  - the phrase "our previous study ..." is overused and repeated in all parts from abstract to discussion - it should not appear in the Results section at all

 - The manuscript's clarity and flow should be  improved by English edition.
